# Intraspecific Comparative Analysis Reveals Genomic Variation of *Didymella arachidicola* and Pathogenicity Factors Potentially Related to Lesion Phenotype

**DOI:** 10.3390/biology12030476

**Published:** 2023-03-21

**Authors:** Shaojian Li, Zhenyu Wang, Meng Gao, Tong Li, Xiaowei Cui, Junhuai Zu, Suling Sang, Wanwan Fan, Haiyan Zhang

**Affiliations:** Institute of Plant Protection, Henan Key Laboratory of Crop Pest Control, International Joint Research Laboratory for Crop Protection of Henan, Key Laboratory of Integrated Pest Management on Crops in Southern Region of North China, Henan Academy of Agricultural Sciences, Zhengzhou 450000, China

**Keywords:** *Didymella arachidicola*, intraspecific, lesion phenotype, genomic variation, pathogenicity factors, comparative analysis

## Abstract

**Simple Summary:**

*Didymella arachidicola*, the causal agent of peanut web blotch, leads to severe defoliation at the late growth stage of peanuts, and eventually to significant yield losses of up to 30%. The biology, ecology, and taxonomy of this phytopathogenic fungi have been well-studied; however, no study has focused on its genomic variation and pathogenic phenotype. Herein, we reported the first chromosome-scale genome assembly of *D. arachidicola*, which provides a reliable baseline for further comparative studies of plant pathogenic fungi. Combined with genome re-sequencing, we revealed the genomic variation within the *D. arachidicola* population, as well as comprehensively analyzed the pathogenicity-related genes to preliminarily explain their roles in forming different lesion phenotypes of peanut web blotch. This work set a genomic foundation and an adaptive landscape of *D. arachidicola* for understanding its genomic diversity and adaptive evolution, in parallel to the correlation of genotype and phenotype underlying the evolutionary force.

**Abstract:**

*Didymella arachidicola* is one of the most important fungal pathogens, causing foliar disease and leading to severe yield losses of peanuts (*Arachis hypogaea* L.) in China. Two main lesion phenotypes of peanut web blotch have been identified as reticulation type (R type) and blotch type (B type). As no satisfactory reference genome is available, the genomic variations and pathogenicity factors of *D. arachidicola* remain to be revealed. In the present study, we collected 41 *D. arachidicola* isolates from 26 geographic locations across China (33 for R type and 8 for B type). The chromosome-scale genome of the most virulent isolate (YY187) was assembled as a reference using PacBio and Hi-C technologies. In addition, we re-sequenced 40 isolates from different sampling sites. Genome-wide alignments showed high similarity among the genomic sequences from the 40 isolates, with an average mapping rate of 97.38%. An average of 3242 SNPs and 315 InDels were identified in the genomic variation analysis, which revealed an intraspecific polymorphism in *D. arachidicola*. The comparative analysis of the most and least virulent isolates generated an integrated gene set containing 512 differential genes. Moreover, 225 genes individually or simultaneously harbored hits in CAZy-base, PHI-base, DFVF, etc. Compared with the R type reference, the differential gene sets from all B type isolates identified 13 shared genes potentially related to lesion phenotype. Our results reveal the intraspecific genomic variation of *D. arachidicola* isolates and pathogenicity factors potentially related to different lesion phenotypes. This work sets a genomic foundation for understanding the mechanisms behind genomic diversity driving different pathogenic phenotypes of *D. arachidicola*.

## 1. Introduction

Peanut web blotch (PWB) is one of the peanut plant’s most yield-limiting foliar diseases caused by *Didymella arachidicola* [1]. It was first observed in Texas, United States, in 1972 [2], then was gradually reported in Liaoning, Shandong, Shaanxi, and Henan provinces of China in the 1980s–1990s [3]. PWB occurs in the middle–late growth periods of peanuts, causes severe defoliation, and results in significant yield losses [3,4]. Due to the economic importance of peanuts, most studies have focused on the biology, ecology, and control methods of PWB, as well as the taxonomy of its pathogens [5,6,7,8]. Based on modern phylogeny, which combines genomics and morphology, the causal fungus of PWB has been recently reclassified as *D*. *arachidicola* [9,10], previously described as *Phoma arachidicola* and *Peyronella arachidicola* [6,11]. However, the mechanisms underlying its pathogenesis and virulence remain to be elucidated.

Phytopathogenic fungi have adopted diverse lifestyles, including obligate biotrophic, hemibiotrophic, and necrotrophic [12,13]. Biotrophic fungi have a narrow host range and derive nutrients from living host cells, while necrotrophic fungi have a broad host range and derive nutrients from dead host cells that are killed by secreted hydrolytic enzymes and toxins. Notably, hemibiotrophic fungi have a narrow host range and can establish an initial biotrophic phase, then switch to a necrotrophic phase [12,14,15]. The shift of fungi lifestyles has been revealed to include switching between the same fungal species with different lifestyles isolated from different host plants [16] or different host cultivars [17,18] and between different fungal species of the same genus isolated from different host plants [19,20]. We have previously found two different leaf lesion phenotypes of PWB in China: reticulation type (R type) and blotch type (B type) (Unpublished work). Isolates of R type invade the epidermal cells and extend over the adaxial leaf surface. Isolates of B type lead to chlorosis or cell death (necrosis), and eventually perform corresponding spot symptoms at both adaxial and abaxial leaf surfaces. Different lifestyles of fungi are highly connected to their secretome (secreted effectors, carbohydrate-active enzymes, transporters, etc.), which are secreted from penetration to tissue destruction [19,20,21]. Comparing variations in the secretome will facilitate our understanding of how fungi interact with their hosts.

Next-generation sequencing technologies have been widely used to study plant pathogens and greatly facilitated understanding of fungal evolution and virulence factors [22,23,24,25,26,27]. The lifestyle of fungal species in Dothideomycetes evolves in four major transitions from non-phytopathogenic to phytopathogenic, along with the pathogenicity-related genes carried in different genera varying considerably [28]. Fungi virulence has been revealed to be associated with the number of certain secreted proteins, including CAZy proteins [29]. A large-scale comparative genomic analysis has been confirmed to be a powerful tool for investigating the genetic variation of phytopathogenic fungi [30,31]. Based on a comparison of 26 *Phytophthora sojae* isolates, a genomic landscape including single-nucleotide polymorphism (SNP), insertion or deletion (InDel), copy number variation (CNV), and core RxLR effectors that are assumed to be essential for fungi infection have been identified [32]. The genomic differences and uncovered gene family expansions in the pathogenicity-related genes have been reported from a comparative genomic analysis of three *Phytophthora capsici* isolates [33]. Although the increasing number of genomic analyses provides insight into elucidating the genetic mechanisms underlying different aggressiveness levels of fungal pathogens, most have focused on the evolutionary traits and genetic variation at the intraspecific levels of plant fungal pathogens [30,33,34,35].

To date, only two draft genome sequences of *D. arachidicola* have been reported that referred to genome sizes of 34.11 and 47.30 Mb [36,37]. Notably, in one of the draft genome reports (our previous study), the causal fungus of PWB was misused as *Peyronellaea arachidicola* [37], which was reclassified as *D. arachidicola* in 2020 [10]. Multiple genomic comparisons at the intraspecific level are believed to reveal the genomic adaptation of *D. arachidicola* and provide a molecular foundation to understand the pathogenic mechanism of *D. arachidicola.* Therefore, this study completed the high-quality, chromosome-scale genome assembly of a local *D. arachidicola* isolate and re-sequenced 40 isolates from 26 geographical locations across China. The present work provides a valuable foundation for understanding the genomic basis of pathogenicity and the adaptive evolution of *D. arachidicola*. Furthermore, our results set a basis for further studies on functional genes related to lesion phenotype.

## 2. Materials and Methods

### 2.1. Fungal Isolates and Pathogenicity Assays

All the isolates of *D. arachidicola* were isolated from symptomatic leaves collected from 26 different geographic locations across main peanut production areas in China (Appendix A). *D. arachidicola* isolates were single spore-derived, and their hyphae (2–3 days old) were soaked in cryogenic vials with sterile 20% glycerin solution for −80 °C storage. For the pathogenicity test, all isolates were transferred to an Oat agar plate medium for induction of conidia production (22 °C in the dark for seven days, then 22 °C under a photoperiod of 12 h black light and 12 h dark). Eight-leaf stage seedlings of peanut cultivar Yuhua 22 were chosen for inoculation. The leaflet of peanut seedlings was inoculated by spraying either spore suspension as treatment or sterile water as a control. Three replicates were set for each treatment. The inoculation seedlings were incubated at 25 °C with a relative humidity of 90% and a photoperiod of 10 h light and 14 h dark. The diseased leaflet was recorded and photographed at 14 days post-inoculation (dpi), blade and lesion areas were measured using ImageJ v1.8.0, and the pathogenicity was evaluated based on a nine-level grading method [38]. The differences in disease index were compared in R software v4.2.2. Bartlett’s test was used to check the homogeneity of variances; the Wilcoxon test was used for unpaired two-sample comparison with *p* values adjusted by the Benjamini method to control the false discovery rate (Figure 1A), and the Mann–Whitney U test was used for comparison of two independent groups (Figure 1B). The standard error of the mean was calculated in Microsoft Office Excel with the formula ‘=STDEV.S (sample)/SQRT (COUNT (sample))’.

### 2.2. Genomic DNA, RNA Extraction, and Whole-Genome Sequencing

All the isolates were grown on Potato Dextrose Agar (PDA) medium for three days at 25 °C with a photoperiod of 8 h light and 16 h dark. Mycelium was harvested, freeze-dried in liquid nitrogen for 3 h, and stored at −80 °C. Genomic DNA and total RNA of *D. arachidicola* isolates were extracted from mycelia using the DNeasy plant mini kit (QIAGEN, Venlo, The Netherlands) and TRI reagent (Sigma Aldrich, St. Louis, MO, USA). The quality of genomic DNA and total RNA was checked using 1% agarose gel electrophoresis and quantified by NanoDrop spectrophotometer 2000c (Thermo Fisher Scitific Inc., Waltham, MA, USA). RNA integrity was accessed using the RNA Nano 6000 Assay Kit of the Agilent Bioanalyzer 2100 system (Agilent Technologies, Santa Clara, CA, USA). De novo genome sequencing was performed on isolate YY187. It was undertaken by producing Single Molecule Real-Time (SMRT) cell libraries and sequenced on the PacBio Sequel platform (PacBio, Menlo Park, CA, USA).

### 2.3. Genome Assembly and Hi-C Analysis

Raw PacBio polymerase reads were processed by SMRT analysis package v2.3.0 to filter out low-quality reads (readScore < 0.8), remove adapters, and extract subreads with a length greater than 1000 bp. The high-quality subreads were corrected and assembled using Canu v1.5 with the parameter ‘correctedErrorRate’ set to 0.045 [39]. Genome completeness and assembly quality were assessed using BUSCO v5.4.4 [40,41].

The procedures of Genomic DNA extraction, quality, and quantity assessment of isolate YY187 were as described above. Hi-C sequencing library was constructed, the concentration and insert size were detected using Qubit v2.0 and Agilent 2100, and then subjected to paired-end 150 bp sequencing by Illumina HiSeq platform (Illumina, San Diego, CA, USA). Illumina clean reads were mapped to the assembled genome of *D. arachidicola* using BWA with default parameters [42]. Valid and invalid interaction pairs of unique mapped read pairs were filtered and assessed using HiC-Pro v2.10 [43]. Scaffolds/contigs were clustered into chromosome groups, and then those scaffolds/contigs within each chromosome group were ordered and oriented using LACHESIS [44] with the parameters ‘-CLUSTER_MAX_LINK_DENSITY 2 -CLUSTER_MIN_RE_SITES 5 -ORDER_MIN_N_RES_IN_TRUNK 5 -ORDER_MIN_N_RES_IN_SHREDS 5 -CLUSTER_NONINFORMATIVE_RATIO 2’.

### 2.4. Genome Annotation

The repeat library was constructed using LTR_FINDER v1.05 [45], MITE-HUNTER [46], RepeatScout v1.0.5 [47], and PILER-DF v2.4 [48], classified using PASTEClassifier [49], and then merged with the Repbase database [50]. Accordingly, all the possible repeat elements (REs) were detected by RepeatMasker v4.0.6 [51].

Protein-coding genes were predicted by combining three strategies: ab initio prediction, homology-based, and RNA-Seq based. The ab initio prediction was carried out by using Genscan v1.0 [52], Augustus v2.4 [53], GlimmerHMM v3.0.4 [54], GeneID v1.4 [55], and SNAP v2006-07-28 [56]. The homology-based prediction was performed by GeMoMa v1.3.1 [57]. RNA-Seq data were mapped to reference transcripts using Hisat2 v2.0.4 and Stringtie v1.2.3 [58]. Transcriptome assembly was conducted based on Unigene sequences that were predicted using PASA v2.0.2 [59] + TransDecoder v2.0 pipeline. The gene sets were integrated into a non-redundant gene set using EVM v1.1.1 [60]. Transfer RNAs (tRNAs) were predicted by tRNAscan-SE [61], and the predictions of ribosome RNAs (rRNAs) and other non-coding RNAs (ncRNAs) were performed using Infernal v1.1 [62] based on Rfam database [63].

The prediction gene set was used for gene annotation based on functional databases of KOG [64], KEGG [65], Swiss-Prot (2015_01), TrEMBL [66], and Nr [67] using BLAST v2.2.29 [68]. The hits from the Nr database blast were further annotated with the Blast2GO v2.5 [69], Hmmer v3.0 [70], GO [71], and Pfam (27.0) [72] database, respectively. In addition, Hmmer v3.0 [70] was used to annotate carbohydrate enzymes, membrane transport proteins, and cytochrome P450 proteins based on the carbohydrate-active enzymes (CAZy) database, transporter classification database (TCDB), and cytochrome P450 engineering database (CYPED), respectively [73,74,75]. The prediction gene set was also used for annotation in the pathogen–host interactions (PHI) database [76] and the database of fungal virulence factors (DFVF) [77] using BLAST v2.2.29 [68]. 

To identify protein subcellular localization, the protein sequences of all the predicted genes were analyzed, and proteins containing signal peptides were detected by SignalP v4.0 [78] with the parameter ‘-f long -g png’. After transmembrane proteins were filtered by TMHMM with default parameters [79], the remains were candidate-secreted proteins. EffectorP [80] was used to further analyze the secreted proteins for predicting effector proteins.

### 2.5. Genome Re-Sequencing and Variations Calling

Genomic DNA extraction of 40 isolates of *D. arachidicola* and quantity/quality determination were performed as described above. Paired-end libraries with an insert size of 350 bp for Illumina sequencing were constructed, and PE150 sequencing was performed on an Illumina HiSeq X-Ten platform. The obtained raw reads were filtered to remove adapters, low-quality reads on which more than 50% of the bases had a quality score less than 20 (Phred-like score), and paired-end reads with >10% ‘N’ bases. The clean reads of re-sequenced isolates were aligned to the reference genome (de novo assembly of isolate YY187) by BWA-MEM v0.7.12 [81]. Duplicate reads were removed from alignments using samtools v1.7 [82]. 

Single-nucleotide polymorphisms (SNPs) and insertions and deletions (InDels) within the 40 isolates were called using the HaplotypeCaller module in GATK [83] and were filtered with the following parameters: QD < 2.0 || MQ < 40.0 || FS > 60.0 || QUAL < 30.0 || MQrankSum < −12.5 || ReadPosRankSum < −8.0 -clusterSize 2 -clusterWindowSize 5. SNP and InDel annotations were performed using default parameters based on the reference genome by snpEff v4.1 [84]. Synonymous and non-synonymous SNPs and InDels in exons were further filtered. In addition, genes with non-synonymous SNPs and InDels in exons were targeted and used for searching genes with potential functional differences, which were further annotated against databases of Nr [67], SwissProt (2015_01), GO [71], COG [64], and KEGG [65] using BLAST v2.2.29 [68].

### 2.6. Population Phylogeny Analysis

The SNPs identified by GATK were further filtered: only SNPs with a minor allele frequency greater than 5% and less than 20% missing data were considered high-quality SNPs. A phylogenetic tree was reconstructed based on the alignment of high-quality SNPs, using the maximum likelihood method with the GTR + G + I model in MEGA X [85] with 1000 bootstrap replicates. The population structure within 40 isolates was inferred based on the high-quality SNPs using ADMIXTURE v1.22 [86], with the putative number of sub-populations (K value) ranging from 1 to 10. The optimal number of sub-populations was assessed using five-fold cross-validation.

### 2.7. Identification of Potential Pathogenicity Factors

Those genes with non-synonymous SNPs and InDels occurring in exons were considered specific differential gene sets for each isolate. As the reference genome we used above was the genome of the most virulent isolate, the differential gene sets of the least virulent isolates of two lesion phenotypes were selected to identify pathogenicity-related genes. The differential gene set was individually compared with the functional annotations of the reference genome, including CAZy-base, TCDB, PHI-base, CYPED, DFVF annotations, and prediction of secreted proteins and effector proteins by VCFtools v.0.1.15 [87]. All the differential gene sets of B type isolates were merged and compared to further validate shared differential genes to validate pathogenicity factors potentially related to different lesion phenotypes.

## 3. Results

### 3.1. Pathogenicity Varied among D. arachidicola Isolates

The results showed differences in the pathogenicity of infected peanut seedlings at 14 dpi. The isolates YY187 and SC291 had the highest and lowest pathogenicities, with a disease index of 28.33 and 7.56, respectively (Figure 1A). The control treatment (seedlings inoculated with sterile water) showed no symptoms during the experimental period. The leaf lesion phenotypes showed at 14 dpi among all the isolates were recorded, which showed that 33 isolates were reticulation type (R type) and 8 were blotch type (B type). The isolates YY187 and SD278 were the most virulent from R and B types, respectively (Figure 1A). In addition, the average disease index of R type isolates was significantly higher than that of B type isolates (Figure 1B).

### 3.2. Genome Assembly and Annotation

Isolate YY187 was sequenced and assembled as a reference genome of *D. arachidicola*. A total of 7.86 Gb filtered subreads with an average length of 8.09 kb were generated from the PacBio Sequel platform. The assembled genome size was 47.35 Mb, comprising 26 scaffolds, with an N50 length of 2.17 Mb and a G + C content of 56.37% (Table 1). A total of 16,629 genes were predicted in this study (Appendix A).

The Hi-C approach was further used to determine the state of genome folding of isolate YY187. Approximately two million paired-end reads (6.01 Gb) were obtained with a G + C content of 51.95% and a Q30 of 94.35%. The ratios of mapped reads and unique mapped read pairs were 96.14% and 73.06%, respectively. Among the unique mapped read pairs, valid interaction pairs (8.83 Mb) with a percentage of 60.32% were generated. Hi-C assembly located 36.03 Mb genomic sequences (accounting for 76.09% of the total sequence length) on chromosomes. Moreover, the chromosomal ordered and oriented sequences accounted for 100% of the total sequence length and number, respectively. The Hi-C assembled results were further cut into 20 Kb bins, the interaction intensity of any two of which was used to construct a heat map. Finally, the preliminary PacBio assembly was assembled using the Hi-C technique into a chromosome-level assembly containing 18 chromosomes (Figure 2, Appendix A).

The predicted genes were annotated based on multiple functional databases (KOG, KEGG, Swiss-Prot, TrEMBL, Nr, GO, and Pfam), with which 15,767 genes (approximately 94.82% of the total) showed at least one hit to the above databases (Appendix A). In total, 9682 protein-coding genes were annotated from the GO database. The ‘cell part’, ‘cell’, ‘membrane’, and ‘organelle’ were the most prevalent GO terms associated with cellular components; meanwhile, the ‘catalytic activity’ and ‘metabolic process’ were the largest categories of genes associated with molecular function and biological processes, respectively (Appendix A). In addition, 7676 protein-coding genes were annotated by the KOG database. The functional category ‘general function prediction only’ was the most, followed by ‘posttranslational modification, protein turnover, chaperones’ and ‘amino acid transport and metabolism’ (Appendix A). The annotation based on the KEGG database found 4739 protein-coding genes, demonstrating that ‘biosynthesis of amino acids’ and ‘carbon metabolism’ were the two categories most associated with metabolism (Appendix A). The density of annotated genes distributed on each chromosome is shown (Figure 2) using gene location visualization in TBtools software for visualization [88].

Additional annotations were based on CAZy-base, TCDB, CYPED, PHI-base, and DFVF. A total of 902 genes were annotated to be CAZymes, including 35.32% glycoside hydrolases (GHs), 19.90% carbohydrate esterases (CEs), 18.00% glycosyl transferases (GTs), 14.02% auxiliary activities (AAs), 9.25% carbohydrate-binding modules (CBMs), and 3.48% polysaccharide lyases (PLs) (Appendix A). A total of 405 and 558 genes were annotated as membrane transport proteins and cytochrome P450 proteins in the reference genome, respectively (Appendix A). A total of 5028 potential pathogenic genes were assigned to different categories in PHI-base (containing information on experimentally proven genes in bacteria, fungi, and oomycetes), most of which were related to reduced virulence (40.93%) and unaffected virulence (30.65%) (Appendix A). As a supplement and validation, 2996 genes were annotated to be known fungal virulence factors in the DFVF database, among which 240 genes were exclusive in the DFVF database as mapping to the annotation genes in PHI-base (Appendix A).

In total, 1906 proteins with signal peptides and 3990 transmembrane proteins were predicted in the genome of *D. arachidicola* isolate YY187. By removing those proteins with a transmembrane structure from proteins with a signal peptide, the remaining 1292 proteins were identified as potentially secreted proteins. Furthermore, 335 secreted proteins (accounting for 25.93% of the total predicted secreted proteins) belonged to CAZymes, among which PHI-base annotated 169 secreted proteins with information of “reduced virulence”, “loss of pathogenicity”, “effector (plant avirulence determinant)”, etc. In total, 144 proteins of the potential CAZymes were supplementarily annotated by DFVF of known fungal virulence factors, with 131 proteins the same to PHI-base annotation. Moreover, 174 effector proteins associated with pathogen–plant host interactions were further predicted by EffectorP by analyzing potentially secreted proteins (Appendix A).

### 3.3. Genomic Variation and Phylogeny of D. arachidicola

Genome re-sequencing of 40 isolates (except for the isolate for reference genome) was performed to investigate the genomic variations of *D. arachidicola* at the intraspecific level. The clean reads of 40 isolates were mapped to the reference genome of isolate YY187, with GC content and mapping rate ranging from 50.95% to 53.03% and 90.13% to 99.66%, respectively (Appendix A). A total of 158,962 SNPs and 12,771 InDels were identified (Appendix A). For all the isolates, the ratio of transition over transversion (Ti/Tv) of SNPs ranged from 2.91 to 3.47, and the insertion and deletion numbers of InDels ranged from 86 to 267 and 101 to 229, respectively (Appendix A). Genomic variations were further categorized due to their occurring regions (intergenic regions, upstream or downstream regions, and exons or introns), which showed that all 40 isolates had similar variations (Appendix A). In addition, most variations (59.62% of SNPs and 59.00% of InDels) were located in gene upstream and downstream regions, and 19.21% of SNPs and 14.03% of InDels were located in exon regions (Figure 3A). Non-synonymous SNPs were filtered in exons with numbers from 224 to 507, and InDels detected in exons ranged from 22 to 62 (Appendix A). An intraspecific comparison of variations among all the isolates is further shown in Appendix A. For the variation types in CDS regions of each isolate, the proportion of synonymous and non-synonymous SNPs was the most; meanwhile, frameshift mutation was the primary type that referred to InDels (Figure 3B). Genes with non-synonymous SNPs and InDels occurring in exons were identified, with the gene numbers ranging from 167 to 420 and 19 to 51, respectively (Appendix A).

A phylogeny analysis was performed to investigate the phylogenetic relationship of the *D. arachidicola* population. The SNP calling was based on the reference assembly of isolate YY187, resulting in 8154 SNPs being further filtered as high-quality SNPs (Appendix A), which were used to construct a maximum likelihood phylogenetic tree (Figure 4). The phylogenetic tree included 40 isolates from 26 geographical regions of China, and the results indicated four sub-populations (S1, S2, S3, and S4), while no correlations were found among the clustering of isolates and geography for the *D. arachidicola* population. Eight isolates of B type belonged to clusters S1, S3, and S4 separately, indicating no correlation between the phenotype of different lesions and geographical regions. In addition, the putative number of sub-populations was assumed from 1 to 10 and assessed using five-fold cross validation. At K = 4, *D. arachidicola* isolates were divided into four groups that showed the same grouping results as our phylogenetic analysis.

### 3.4. Identification of Pathogenicity Factors

According to the results of pathogenicity assays, the most virulent isolate was YY187, the least virulent R type isolate was SC289, and the B type was SC291. Based on the genome re-sequencing, 3,552,798 clean reads of SC289 and 3,510,557 clean reads of isolate SC291 were mapped to the reference genome (isolate YY187) with a mapping rate of 98.36% and 98.70%, respectively (Appendix A). By variation calling and annotations, 315 and 420 differential genes with non-synonymous SNPs, and 43 and 41 differential genes with InDels, were further annotated (Appendix A). Differential genes were blasted in general functional databases that generated two differential gene sets containing 266 and 331 functional annotated genes for isolates SC289 and SC291, respectively (Appendix A). The two differential gene sets were further integrated into one gene set containing 512 differential genes (85 genes were shared by two differential gene sets), which we called an integrated gene set (Appendix A) and used for subsequent analysis.

Compared with the reference genome annotations in CAZy-base, TCDB, CYPED, PHI-base, and DFVF, the pathogenicity-related genes were further identified from 512 genes from an integrated gene set, which were 35 CAZyme genes, one membrane transport protein gene, 29 cytochrome P450 monooxygenase genes, 193 pathogen–host interaction protein genes, and 126 known fungal virulence factors (Appendix A). It has been well-understood that plant pathogens secrete effector proteins to facilitate the infection of their hosts [89,90,91]. By comparing the integrated gene set to the predicted secreted proteins of the reference genome, 30 genes were identified as secreted proteins, among which three were further identified as effector proteins (Appendix A).

### 3.5. Pathogenicity Factors Potentially Related to Different Lesion Phenotypes

Compared with the reference isolate YY187 (R type), eight B type isolates, SD278, HuB260, SX282, SX283, LN297, SC290, SC291, and SC293, were analyzed to identify the potential pathogenicity factors related to different lesion phenotypes. In total, 13 differential genes were shared by all B type isolates (Table 2). Those genes were mainly located in chromosomes 3 and 11, with only one in Chr1 (Figure 5). Eight genes had hits in PHI-base with either “reduced virulence”, “unaffected pathogenicity”, or “loss of pathogenicity” annotations, five of which also had hits in DFVF annotations (Table 2). Only one gene in Chr1 was also annotated in CYPED. One out of three genes in Chr3 was annotated to be CAZyme. On Chr11, two secreted proteins were found, and one of them (EVM0007819.1) was further predicted to be an effector protein (244 amino acids).

## 4. Discussion

Inoculation of *D. arachidicola* isolates on peanut seedlings showed that pathogenicity varied among 41 isolates from different regions of China (Figure 1A). Given the two different lesion phenotypes, isolates of reticulation type (R type) and blotch type (B type) showed significant differences in pathogenicity (Figure 1B). Notably, the lesion phenotype exhibited no correlation with geographic locations. Different lesion phenotypes can be found in different regions, such as isolate SD277 and SD279 (R type) and SD278 (B type), originating from Shandong province, China. Previous studies have reported that the adaptability and pathogenicity of fungal pathogens can be affected by plant cultivars and environmental factors such as temperature, light conditions, and humidity [34]. Pathogenic fungal species with a narrow range of plant species were so-called ‘host specificity’ [92] and have been extensively used to study pathogen–host interactions [93,94,95]. It has been proved that fungal pathogens utilize diverse life strategies to interact with host plants, while plants use immune systems to respond to pathogenic infection [12,96]. In order to colonize host plants, fungal pathogens either directly invade epidermal cells, or extend hyphae over, between, or through plant cells, which happens to be the different lesion phenotypes in our study (Figure 1C). In consideration of the limited expansion and higher invasion ability of B type isolates, a better strategy for interfering with PAMP (pathogen-associated molecular patterns)-triggered immunity (PTI) and further effector-triggered susceptibility (ETS) might happen. However, the causation of different lesion phenotypes of *D. arachidicola*, especially at the genetic level, needs to be further illustrated.

The results of genome re-sequencing showed a high mapping rate of the clean reads mapped to the reference, around 97~99%, except some lower, around 92~94% (Appendix A). The present bioinformatic methods usually discarded those unmapped reads from the analysis process, while some studies revealed that meaningful biological information (e.g., missing genes) could also be found by further exploring unmapped reads [97,98]. However, this phenomenon was rarely reported in the research on fungal pathogens. Considering the high homology shown in *D. arachidicola* isolates at the intraspecific level, the lower mapping rate may be more related to the sequencing quality than sequence differences. Nevertheless, a comparison of PacBio whole-genome assemblies of representative isolates with the reference could better explain the unmapped reads issue and needs further investigation. The results of the variation calling indicated a homogeneous polymorphism at the intraspecific level of *D. arachidicola*. The 40 isolates of *D. arachidicola* showed similar numbers of SNPs and InDels; moreover, the proportion of variations identified, respectively, in the intron, exon, and other genomic regions showed no significant differences among isolates. Regarding the distribution in genomic regions, genomic variations were mainly located in the upstream and downstream regions (59.31%) and exon regions (16.62%) of genes, which showed a similar distribution pattern with variations of three pathotypes of a rust fungi *Puccinia striiformis* [99], but differed from the previous study of *Phytophthora sojae* in that a high proportion of SNPs and InDels were located in intergenic regions [32]. The upstream and downstream regions have been mainly studied at the cis-regulatory elements, such as promoters and enhancers, which have been elucidated to regulate gene expression and functions [100,101]. Thus, most of the variations identified in our study were distributed in the non-coding regions that might contribute to phenotypic diversity within species by affecting the functions of cis-regulatory elements. In the coding regions, especially exon regions, the variations are directly related to the transcription and translation of proteins. The phylogenetic relationship based on SNPs did not correlate with their geographical origins and virulence, suggesting a more diverse evolution at the intraspecific level of the *D. arachidicola* population. As the latitudes of origins, peanut cultivars, and environmental factors can affect the adaptability of *D. arachidicola* isolates, an extensive range of sample collection and further investigations combined with the related environmental factors will contribute to a better understanding of the evolution of *D. arachidicola*.

By comparing the differential gene sets of all B type isolates, 13 shared differential genes were identified as pathogenicity factors potentially related to different lesion phenotypes of *D. arachidicola* (Table 2). Among these genes, EVM0007100.1 had hits in PHI-base and DFVF simultaneously with the name of *GAS1*, which has been reported to be essential for efficient colonization on the host surface [102] by affecting the formation of appressoria and penetration into the epidermal cell layer [103]. As homologous to *GAS1*, EVM0007100.1 is annotated as CAZyme with GH31 (glycoside hydrolase) activity. Previous studies confirmed that various CAZymes had been found to be important virulence factors, among which glycoside hydrolases (GHs) are the most abundant class studied so far [104]. The known GHs families in phytopathogenic fungi and oomycetes that were reported to have potential roles in penetration of the cell wall, expansion of the fungus in the host, stealth pathogenicity, et cetera, were concentrated in GH5-7, GH10-13, GH18, GH45, and GH74 families [105]. Here, we showed the potential that members from the GH31 family could also be regarded as CAZymes with potential functions on pathogen penetration and invasion in phytopathogenic fungi.

Earlier studies have reported that plant pathogens utilize secreted proteins termed ‘effectors’, a broad class of cytotoxic, virulence-promoting, or resistance-eliciting molecules released from pathogen cells to facilitate infection by suppressing plant defense reactions and manipulating plant cell physiology [12,91]. The gene EVM0007819.1 identified in this work was predicted to be an effector protein and was functionally annotated as necrosis-and ethylene-inducing protein 1 (Nep1), which belongs to Nep1-like proteins (NLPs). NLPs are widespread among microbes such as bacteria, fungi, and oomycetes and act as toxin-like virulence factors that induce tissue necrosis [106,107]. Studies have proved that plant pathogens generally expressed multiple NLPs with necrotrophic or hemibiotrophic lifestyles during infection or in the transition from biotrophic to necrotrophic [108,109]. EVM0015380.1 was functionally annotated as a *Kpp6* gene, which has been identified for encoding a mitogen-activated protein (MAP) kinase [110]. As fungal MAP kinases have shown essential roles in controlling essential virulence factors, the *Kpp6* gene has been further speculated to be crucial for the penetration of plant epidermis [110,111]. In addition, the genes EVM0000257.1, EVM0000544.1, EVM0002645.1, and EVM0013695.1 were annotated against PHI-base to be putative transcription factors (TFs) [112], which are essential regulators of gene expression, possess critical roles in the signal transduction pathways, and further phenotypic evolution [113,114].

## 5. Conclusions

This work assembled the chromosome-scale genome of a local isolate of *D. arachidicola* as the first reference and re-sequenced the genomes of 40 *D. arachidicola* isolates from 26 geographical locations across China. Our analysis revealed that the pathogenicity varied among all the isolates, and the isolates of B type presented less virulence than the R type in consideration of the different lesion phenotypes, although there were no correlations among pathogenicity, lesion phenotype, and geographical origins. Furthermore, we investigated the genomic variations of *D. arachidicola* and determined an intraspecific polymorphism. The 13 differential genes potentially related to lesion phenotype were identified to further understand the correlation of genotype and phenotype underlying the pathogenic mechanism. Our study set a genomic foundation for the adaptive evolution of *D. arachidicola* and provided a molecular background for further functional study on the genes potentially related to different lesion phenotypes.

## Figures and Tables

**Figure 1 biology-12-00476-f001:**
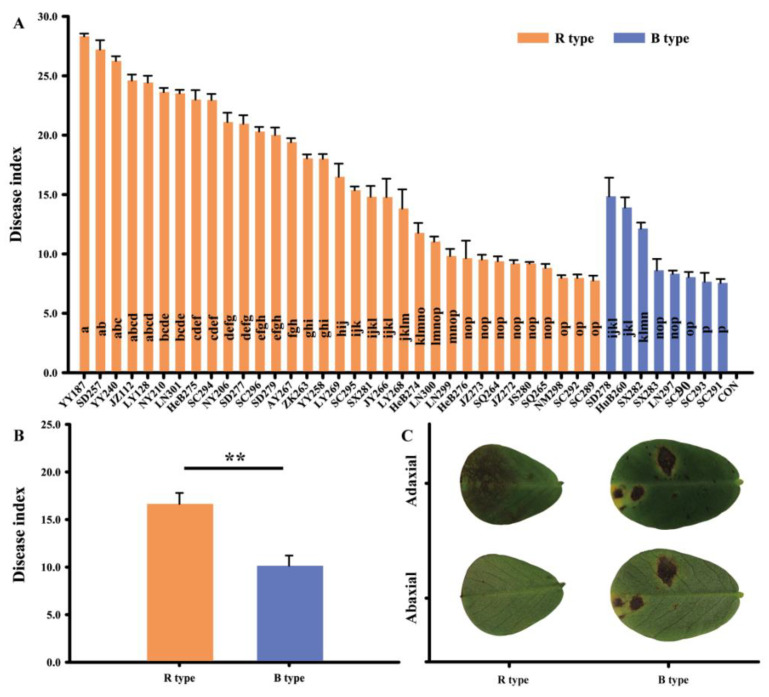
Pathogenicity assays and lesion phenotype of 41 *D. arachidicola* isolates. (**A**) The disease index of peanut seedlings infected by 41 *D. arachidicola* isolates at 14 dpi; CON denotes non-infected peanut seedlings. Letters on the bars denote significant differences in disease index at 0.05 level; error bars denote the standard error of the mean. (**B**) The average disease index of peanut seedlings infected by isolates of different lesion phenotypes at 14 dpi. “**” denotes significant differences at 0.05 level; error bars denote the standard error of the mean. (**C**) Adaxial and abaxial leaf surfaces of different lesion phenotypes showed at the top and bottom, respectively.

**Figure 2 biology-12-00476-f002:**
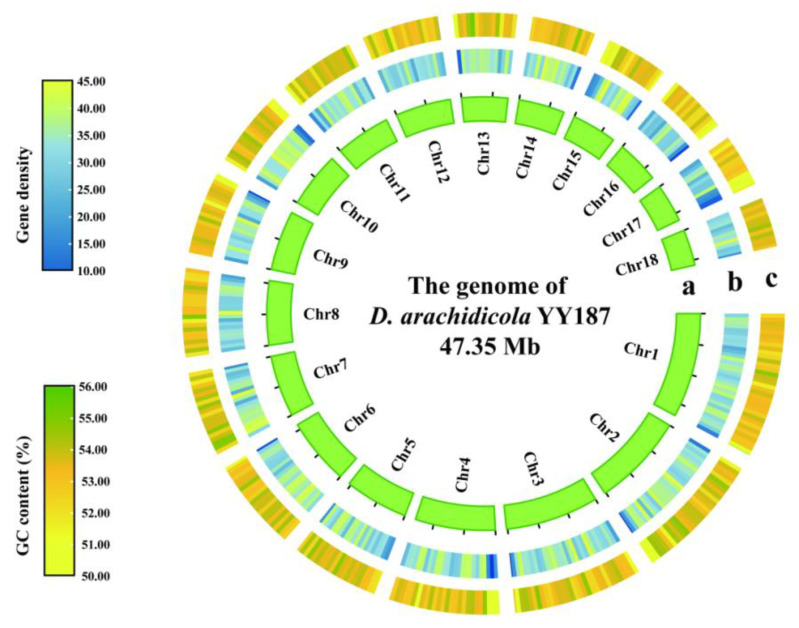
The genomic circos map of *D. arachidicola* isolate YY187. Circles from inside to outside are (a) chromosome information of the *D. arachidicola* genome (scale marks in Mb). (b) Gene density on different chromosomes, color-coded from blue to yellow, represents values from low to high. (c) GC content, color-coded from yellow to green, represents values from low to high. Visualization was performed using advanced circos in TBtools v1.098769.

**Figure 3 biology-12-00476-f003:**
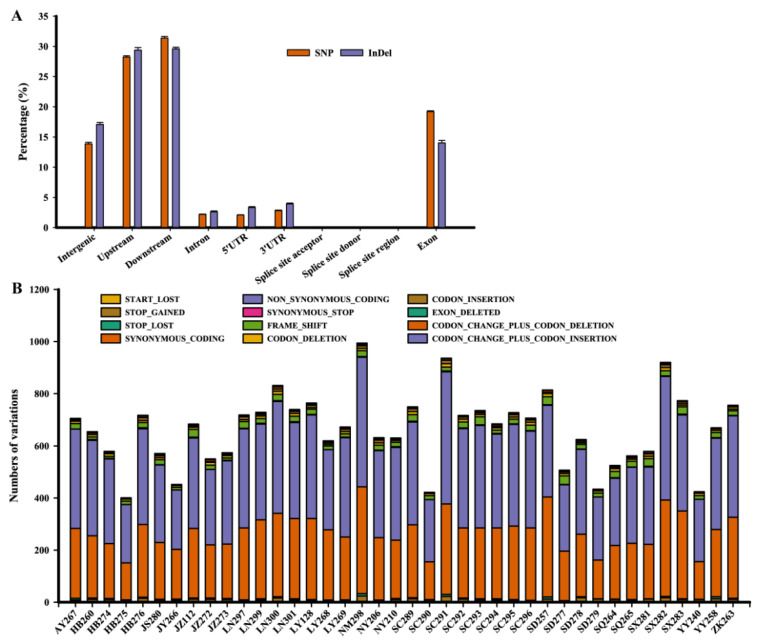
Genome-wide distribution and types of variations (identified by comparison with the reference genome of isolate YY187) within all *D. arachidicola* isolates. (**A**) Genome-wide percentage distribution of SNPs and InDels within different genomic regions, respectively. Each bar indicates the average percentage of variations in a certain genomic region among all the isolates, with the error bars showing the standard error of the mean; (**B**) the proportion of different variation types in the CDS region of *D. arachidicola* isolates.

**Figure 4 biology-12-00476-f004:**
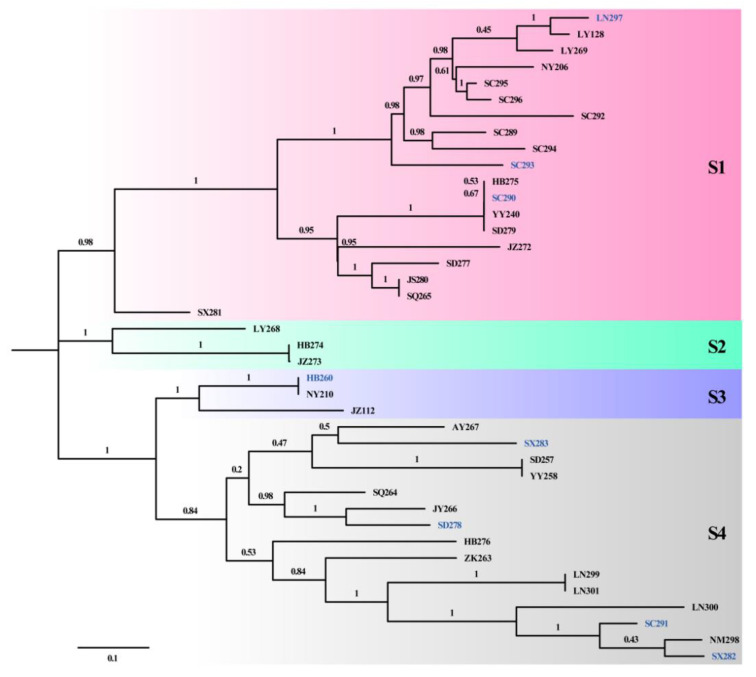
Phylogenetic analysis of *D. arachidicola* population with high-quality SNPs. The SNPs of 40 isolates were called by GATK based on the fact that the genome of isolate YY187 was selected as the reference, which caused the absence of isolate YY187 on the phylogenetic tree. The maximum likelihood (ML) tree was constructed using MEGA X with 1000 bootstrap replicates. The tree was visualized by modification in FigTree v1.4.4. Bootstrap values are shown above the branches. The sub-populations of *D. arachidicola* were classified and displayed by different colors. The isolates of B type are designated in blue font color.

**Figure 5 biology-12-00476-f005:**
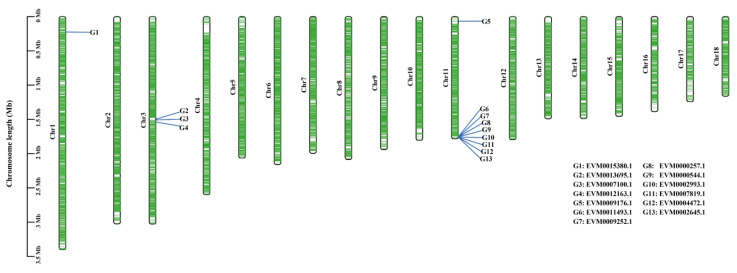
Chromosomal distribution of 13 differential genes. The green lines represent gene distribution on different chromosomes. G1–G13 refer to the 13 genes and mark the location of genes on different chromosomes using a blue line. TBtools v1.098769 visualized the figure.

**Table 1 biology-12-00476-t001:** Summary of genome assembly and annotation in isolate YY187.

Feature	Value
Scaffold number	26
Total scaffold length (Mb)	47.35
Scaffold N50 (Mb)	2.17
Scaffold N90 (Mb)	1.39
Maximum scaffold length (Mb)	6.56
GC content (%)	56.37
Chromosomal located sequence length (Mb)	36.03
Number of chromosomes	18
Percentage of repeats (%)	9.89
Number of predicted genes	16,629
Average gene length (Kb)	1.99
Average CDS number	2.47
Number of non-coding RNAs	tRNA (286), rRNA (81), other ncRNA (95)

**Table 2 biology-12-00476-t002:** Summary of pathogenicity factors potentially related to different lesion phenotypes.

Gene ID	Brief Description	GO ID	Pfam Annotation
EVM0000257.1 ^a^	--	GO:0000981; GO:0003677; GO:0005634; GO:0006351; GO:0006355; GO:0008270	Fungal Zn(2)-Cys(6) binuclear cluster domain
EVM0000544.1 ^a^	Hypothetical protein	GO:0003700; GO:0008270	AAA domain
EVM0002645.1 ^a, b^	Carminomycin 4-O-methyltransferase	GO:0008171	O-methyltransferase domain
EVM0002993.1	--	GO:0003676	Zinc-finger double domain
EVM0004472.1 ^a, b^	3-oxoacyl-[acyl-carrier-protein] reductase	GO:0004316; GO:0006633; GO:0051287; GO:0102131; GO:0102132	Enoyl-(Acyl carrier protein) reductase
EVM0007100.1 ^a, b, c^	Alpha-glucosidase	GO:0004553; GO:0005975; GO:0030246	Glycosyl hydrolases family 31
EVM0007819.1 ^a, b, f^	NPP1 family protein	-	Necrosis inducing protein (NPP1)
EVM0009176.1	Isopenicillin N synthase family oxygenase	GO:0016491; GO:0046872; GO:0055114	non-haem dioxygenase in morphine synthesis N-terminal
EVM0009252.1 ^e^	Agmatinase	GO:0016813; GO:0046872	Arginase family
EVM0011493.1	Hypothetical protein	-	Heterokaryon incompatibility protein Het-C
EVM0012163.1	Hypothetical protein	GO:0004843; GO:0006511; GO:0016579; GO:0036459	Ubiquitin carboxyl-terminal hydrolase
EVM0013695.1 ^a^	AAA family ATPase	GO:0005524	ATPase family associated with various cellular activities (AAA)
EVM0015380.1 ^a, b, d^	Hypothetical protein	GO:0004672; GO:0005524; GO:0006468	Protein kinase domain

^a, b, c, d^ denote the proteins with PHI-base, DFVF, CAZy-base, and CYPED hits, respectively. ^e^ denotes the secreted protein. ^f^ denotes the effector protein.

## Data Availability

The whole-genome project of *D. arachidicola* was deposited at DDBJ/ENA/GenBank under Bioproject PRJNA562378 and the accession numbers can be found in the article/Appendix A.

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
