# Peer review of "Intraspecific Comparative Analysis Reveals Genomic Variation of Didymella arachidicola and Pathogenicity Factors Potentially Related to Lesion Phenotype"

_biology, 2023, doi:10.3390/biology12030476_

Round 1

Reviewer 1 Report

This paper deals with genomic variations of the fungal pathogens D. arachidicola among 40 isolates and pathogenicity factors. Some minor corrections are needed before acceptance : 

On the whole, each Figure contains too much information in a small size and Figures are difficult to read properly. Especially Figure 2 and Figure 3 should be separated in several figures for a better understanding. 

In Figure 1B, the standard deviation seems to be low compared to the value presented in Figure 1A. Disease index for R type isolates goes from 10 to 30 ans the standard deviation. Explanation should be given on how the standard deviation was calculated and how the significant difference at 0,01 level was obtained. 

Author Response

This paper deals with genomic variations of the fungal pathogens D. arachidicola among 40 isolates and pathogenicity factors. Some minor corrections are needed before acceptance:

Point 1: On the whole, each Figure contains too much information in a small size and Figures are difficult to read properly. Especially Figure 2 and Figure 3 should be separated in several figures for a better understanding.

Response 1: We apologize that our original Figures did not show the information properly. We have tried to improve them by providing TIFF versions in the high resolution and re-organized Figure 2 and Figure 3 by moving some of sub-figures into supplementary materials. We hope that they are now understandable and clear enough.

Point 2: In Figure 1B, the standard deviation seems to be low compared to the value presented in Figure 1A. Disease index for R type isolates goes from 10 to 30 ans the standard deviation. Explanation should be given on how the standard deviation was calculated and how the significant difference at 0,01 level was obtained.

Response 2: Thanks, this is very important. In both Figure1A&B, the error bars we labeled were based on the calculation of standard error of the mean (SE), by which we think could better estimate the variability of disease index across isolates of D. arachidicola population in Figure 1A and between different lesion phenotypes in Figure 1B, respectively. The sample size of groups in Figure 1B was greater than that in Figure 1A (three replicates per isolate), we think this is why the standard error shows low compared to the value presented in Figure 1A. We apologize for not specifying the standard error in this work. We have re-analyzed our data and added more details about the calculation of standard error in section 2.1 and the statement on SE used in the legend of Figure 1.

Reviewer 2 Report

The authors present the first chromosome-scale genome assembly of a local isolate of D. arachidicola and re-sequence the whole genomes of 40 isolates from 26 locations across China. Their analysis revealed a high degree of genomic variation among the isolates, which was accompanied by differences in pathogenicity, with isolates of B type being less virulent than those of R type. The authors found 13 genes that could be associated with differences in lesion phenotype. This study provides a foundation for the understanding of the adaptive evolution of D. arachidicola and could lead to further functional studies on genes associated with lesion phenotype. Overall, I believe that this study provides a valuable contribution to our understanding of the genomic basis of pathogenicity in D. arachidicola and will be of great interest to researchers in the field of plant pathology.

One potential limitation of the study is the limited number of isolates used. Although the authors included isolates from 26 geographical locations across China, the sample size is still relatively small, and it would be useful to expand the analysis to include more isolates from different regions to gain a more comprehensive understanding of the genomic variations and pathogenicity factors of D. arachidicola. Additionally, the functional validation of the identified differential genes would strengthen the conclusions of the study.

Figure 1: the statistical analysis should be further described, including the specific hypothesis being tested and the assumptions made for the statistical tests used.

The results of the pathogenicity assay should be compared with previous studies, if available, to assess the reliability of the results obtained in this study.

The use of multiple tools for gene prediction, followed by integration into a non-redundant set, is a common approach, but it would be good to have a comparison of the results obtained from each tool to see the contribution of each tool to the final gene set.

For the gene annotation process, it would be useful to see a summary of the number of genes annotated to each functional database and the functional categories that are overrepresented or underrepresented.

The method for detecting signal peptides and transmembrane proteins should be described in more detail, including the parameters used and how the predictions were validated.

Author Response

Point 1: One potential limitation of the study is the limited number of isolates used. Although the authors included isolates from 26 geographical locations across China, the sample size is still relatively small, and it would be useful to expand the analysis to include more isolates from different regions to gain a more comprehensive understanding of the genomic variations and pathogenicity factors of D. arachidicola. Additionally, the functional validation of the identified differential genes would strengthen the conclusions of the study.

Response 1: We understand the reviewer’s concern. We agree that more sample sites and a larger sample size would be useful for expanded analysis and a better understanding of the genomic variations. However, our sampling sites spread over the main peanut producing areas of China, we think that 41 isolates from 26 geographical locations used in our study are capable of analyzing the genomic variations of Didymella arachidicola.

The reviewer’s suggestion for more samples and the functional validation of those identified differential genes are insightful and constructive, providing a good direction for our further research.

Point 2: Figure 1: the statistical analysis should be further described, including the specific hypothesis being tested and the assumptions made for the statistical tests used.

Response 2: Thanks for pointing this out. We have re-analyzed our data and added details about the statistical analysis in section 2.1.

Point 3: The results of the pathogenicity assay should be compared with previous studies, if available, to assess the reliability of the results obtained in this study.

Response 3: This is a helpful suggestion. In the pathogenicity part, we focus on the evaluation of pathogenicity differences between isolates with different lesion phenotypes. For the causal fungus of peanut web blotch, Didymella arachidicola, there are no previous studies conduct research on the pathogenicity differences correlated with lesion phenotypes. Moreover, a reasonable number of replicates are set during the pathogenicity assay, we believe that makes our data solid for further evaluation.

Point 4: The use of multiple tools for gene prediction, followed by integration into a non-redundant set, is a common approach, but it would be good to have a comparison of the results obtained from each tool to see the contribution of each tool to the final gene set.

Response 4: This is a great suggestion, we have provided a table about the statistics of gene prediction results obtained from each tool as supplementary material. Please see Supplementary Table S1.

Point 5: For the gene annotation process, it would be useful to see a summary of the number of genes annotated to each functional database and the functional categories that are overrepresented or underrepresented.

Response 5: This is important. We have implemented the reviewer’s suggestion about the functional categories by adding additional information in section 3.2. In support of these statements, we provided a summary of the gene numbers annotated from each functional database and the results of all database annotation, which have been integrated into one table as Supplementary Table S2. We also have provided additional Figures about the functional categories of the GO, KOG, and KEGG annotations as Supplementary FigureS3, Figure S4, and Figure S5, respectively.

Point 6: The method for detecting signal peptides and transmembrane proteins should be described in more detail, including the parameters used and how the predictions were validated.

Response 6: Thanks, this is a great suggestion, we have added more info by providing the parameters (SignalP v4.0 with the parameter ‘-f long -g png’; TMHMM with default parameters) for detecting signal peptides and transmembrane proteins in section 2.4.

Reviewer 3 Report

In the manuscript under evaluation, the authors applied intraspecific comparative analysis to reveal the genomic variation and pathogenicity factors of Didymella arachidicola potentially related to lesion phenotype. Some correction and explanations; however, should be made before publication of the manuscript. The following are the major comments. Other minor comments and suggestions could be found in the attached PDF file.

1. Several sentences are not clear. I strongly recommend that the manuscript be revised by a language editing service or a native speaker.

2. More information on similar previous studies on other fungal species is lacking in the introduction section.

3. Why didn’t the authors check the RNA integrity (RIN)? What was the quality of the isolated RNA and how did the authors examine the quality?

4. How many SNPs were detected and how many (of them) were filtered as high quality and used for the phylogenetic analysis? Please specify these numbers in section 2.6.

5. A list of all identified SNPs and indels should be provided along with a list of those used for phylogenetic analysis.

6. Figures’ quality is very low.

7. Were the 13 genes suggested to be lesion phenotype-specific genes found in the other R type isolates (except YY187) and didn't show any variation among all these isolates?

8. It would be interesting if the authors performed a population structure analysis to examine the relationship between geographical distribution and genetic diversity.

9. The authors mentioned GHs gene families, especially GH31, and their roles as virulence factors without relating this information to the obtained results. Overall, the discussion needs significant improvements.

10. Had the authors published the assembled sequence along with annotation on the NCBI database. I was not able to download the whole assembly and annotation. Only individual chromosome sequences are there. Moreover, the unplaced sequences are not published.

11. Did all those genes mentioned in the discussion have variation among studied isolates? If so, please discuss this in detail. and If no, please discuss the possible reason behind the lack of variation, nevertheless you have found a great variation in pathogenicity of different isolates.

Author Response

The reviewer’s comments are insightful and constructive, which can help us improve the manuscript considerably. We have revised our manuscript following the reviewer’s major comments and further modified the paper following the minor comments in the attached PDF file.

Point 1: Several sentences are not clear. I strongly recommend that the manuscript be revised by a language editing service or a native speaker.

Response 1: Thanks, we have implemented the reviewer’s suggestion by revising those unclear sentences and asked our colleagues who are native speakers to polish our writing.

Point 2: More information on similar previous studies on other fungal species is lacking in the introduction section.

Response 2: Thanks, this is a helpful suggestion. We have added previous studies on other fungal species about the investigation of genomic differences, identification of pathogenicity-related genes, and phylogenomic comparisons of fungal isolates etc. in the Introduction section.

Point 3: Why didn’t the authors check the RNA integrity (RIN)? What was the quality of the isolated RNA and how did the authors examine the quality?

Response 3: We appreciate the reviewer bringing this up. We assessed the RNA integrity based on the RNA integrity number and 28S/18S ratio before our further analysis, and have added the description on the check of RNA integrity in section 2.2.

Point 4: How many SNPs were detected and how many (of them) were filtered as high quality and used for the phylogenetic analysis? Please specify these numbers in section 2.6.

Response 4: Thanks, this is a helpful observation. In our study, there are 158,962 SNPs detected, and 8,154 SNPs were filtered as high-quality and used for the phylogenetic analysis. We have added these specific numbers as necessary results in section 3.3.

Point 5: A list of all identified SNPs and indels should be provided along with a list of those used for phylogenetic analysis.

Response 5: Agreed, this is important. We have provided the lists of all identified SNPs, InDels, and the high-quality SNPs used for phylogenetic analysis as Supplementary Table S4.

Point 6: Figures’ quality is very low.

Response 6: This is a helpful observation, and so we have tried to improve them by re-organizing some of the Figures and providing TIFF versions in high resolution.

Point 7: Were the 13 genes suggested to be lesion phenotype-specific genes found in the other R type isolates (except YY187) and didn't show any variation among all these isolates?

Response 7: Thanks for pointing this out. Our analysis aims to find out the pathogenicity factors potentially related to different lesion phenotypes, we thus think the comparisons from isolates with different lesion phenotypes can bring in extra interference due to the effects of different latitudes of origins, peanut cultivars or environmental factors. As the differential gene sets for each isolate were annotated based on the reference (complete genome of isolate YY187, R type), we think a comparison of differential gene sets from all B type isolates could be more reasonable. We do appreciate the reviewer’s concern. We have corrected the statement of ‘lesion phenotype-specific genes’ to avoid misunderstanding and will take full account of the variations among all the isolates by further performing GWAS analysis.

Point 8: It would be interesting if the authors performed a population structure analysis to examine the relationship between geographical distribution and genetic diversity.

Response 8: This is a helpful suggestion. We performed a population structure analysis as the supplementary for the sub-population division of our phylogeny analysis. It aims to investigate the correlation between isolates with different lesion phenotypes and geographical regions. Based on the genome re-sequencing, our study focuses on the genomic variation among the isolates with different lesion phenotypes. Nevertheless, the reviewer’s suggestion are very insightful and constructive, and we will conduct the population structure analysis in our future comparative genomics analysis.

Point 9: The authors mentioned GHs gene families, especially GH31, and their roles as virulence factors without relating this information to the obtained results. Overall, the discussion needs significant improvements.

Response 9: Thanks, this is important. We have added additional information for the correlation between our obtained results and GH31 activity in the corresponding part of the Discussion. Besides, we have checked through and tried to improve our Discussion to make our statement precisely.

Point 10: Had the authors published the assembled sequence along with annotation on the NCBI database. I was not able to download the whole assembly and annotation. Only individual chromosome sequences are there. Moreover, the unplaced sequences are not published.

Response 10: Thanks for pointing this out. We have submitted to and released our genome assembly on NCBI with the Accession Number GCA_016630955.2. However, during this round of revision, we found three unplaced contigs were auto-trimmed by Genbank from our genomic fasta. Since we are sure that these contigs are from our fungi sequencing but not the contaminated sequences, moreover, our annotation is also based on this complete assembly, we are now trying to fix this on Genbank as well as submit our relevant annotation file. This fixation on Genbank is under processing, and we will add this info once we get the new version of the Accession Number of our genomic data. To resubmit our revision with the necessary documents, we provide the whole assembly and annotation files on a shared google doc (https://drive.google.com/drive/folders/19ltP3Wiok5YfArpClRr_Roky1aKGDtEa) Please feel free to download them for your convenience.

Point 11: Did all those genes mentioned in the discussion have variation among studied isolates? If so, please discuss this in detail. and If no, please discuss the possible reason behind the lack of variation, nevertheless you have found a great variation in pathogenicity of different isolates.

Response 11: We understand the reviewer’s concern. The genes in the Discussion are those shared differential genes by all B type isolates, which we consider pathogenicity factors potentially related to lesion phenotype. The variations related to those differential genes may be present or absent in other R type isolates due to the effect of latitude of origins, peanut cultivars, or environmental factors. We think the comparison between B and R type will bring in extra interference for our identification. Thus in our present study, a comparison of differential gene sets only from all B type isolates is comprehensive and reasonable.

Reviewer 4 Report

The manuscript by Li et al. presents a new reference genomic sequence of high quality for a virulent strain of Didymella arachidicola, the causal agent of peanut web blotch. Resequencing and variant calling was performed using an additional 40 D. arachidicola isolates from 26 geographic locations across China.

The genomic and bioinformatic methods in the paper are generally sound using contemporary cost-effective tools and workflows. However, the approach has been designed to minimize the observation of variation across the 41 genomes. Fungi and oomycetes have a tendency to have local clusters of genes responsible for secondary metabolism and for their pathogenicity (e.g., https://doi.org/10.1016/j.gde.2019.07.006). And within a species there is evidence that some of these clusters or individual genes of importance get deleted or amplified (e.g., https://doi.org/10.3389/fmicb.2020.581698). This study is blind to all this unnecessarily. It can only see SNPs and small-indel variants. Pathogenesis-gene duplications, gene fusions and novel horizontally-transferred genes will all be missed.

The authors should go back to the reads from each isolate that failed to map to the reference genome and assemble them, separately for each isolate. This is 7% of all the reads of some isolates (e.g., LY268 in Supplemental Table S2). Then they should identify what genes they find in these contigs. This will identify genes in some of the isolates that are missing in their reference. To find genes in their reference missing from individual isolates, they should look at the mapped read depth for each isolate against the reference. Whole reference genes that have 0 read depth for some isolates but not others are indicative of possible gene loss in that isolate relative to the reference. (Of course there is bias in this suggestion as those isolates that are highly diverged from the reference will map more poorly than those that are very close evolutionarily. However no mapping at all across an entire gene while other isolates do map is certainly suggestive.)

A more complete picture of the structural variants present, particularly the tandem duplicated gene clusters that no doubt exist, could have been obtained by doing full PacBio assembly on several strategically-chosen isolates. Using short-read sequencing there are approaches to look at read depth variation, split-reads and the like to get an estimate of the structural variation present. But these approaches yield many false positives and are never as effective as long-read approaches.

I think this paper is a nice work overlll, but I would very much like to see an analysis of the assembled unmapped reads for the isolates, and a paragraph about what this study has likely missed by not doing at least 2 full PacBio assemblies in the Discussion before it is published.

Minor issue:
Supplemental Table S2: In the column 'Clean Reads', the commas in the numbers are in the wrong places for HuB260, LY128, NY206, SD257 making the numbers look huge at first glance.

Author Response

Point 1: The genomic and bioinformatic methods in the paper are generally sound using contemporary cost-effective tools and workflows. However, the approach has been designed to minimize the observation of variation across the 41 genomes. Fungi and oomycetes have a tendency to have local clusters of genes responsible for secondary metabolism and for their pathogenicity (e.g., https://doi.org/10.1016/j.gde.2019.07.006). And within a species there is evidence that some of these clusters or individual genes of importance get deleted or amplified (e.g., https://doi.org/10.3389/fmicb.2020.581698). This study is blind to all this unnecessarily. It can only see SNPs and small-indel variants. Pathogenesis-gene duplications, gene fusions and novel horizontally-transferred genes will all be missed.

Response 1: Thanks, this is a very constructive suggestion. However, this phenomenon has yet to be reported in the fungus Didymella arachidicola. This study focuses on the genomic variation among the isolates and pathogenicity factors potentially related to different lesion phenotypes based on the genome re-sequencing. We think the data will provide a potential database to identify the pathogenesis-related genes, and our results set a foundation for further functional study.

We thank the reviewer for the kind suggestion and for providing the relevant literature, which provides us with some great ideas on our further comparative genomics, metabonomics, and chemical analysis, e.g., identification of metabolic gene clusters (MGCs), the correlations between MGCs, and fungal diversity, etc.

Point 2: The authors should go back to the reads from each isolate that failed to map to the reference genome and assemble them, separately for each isolate. This is 7% of all the reads of some isolates (e.g., LY268 in Supplemental Table S2). Then they should identify what genes they find in these contigs. This will identify genes in some of the isolates that are missing in their reference. To find genes in their reference missing from individual isolates, they should look at the mapped read depth for each isolate against the reference. Whole reference genes that have 0 read depth for some isolates but not others are indicative of possible gene loss in that isolate relative to the reference. (Of course there is bias in this suggestion as those isolates that are highly diverged from the reference will map more poorly than those that are very close evolutionarily. However no mapping at all across an entire gene while other isolates do map is certainly suggestive.)

Response 2: We appreciate your suggestion. However, our analysis was based on re-sequencing genomics rather than whole-genome sequencing. Therefore the unmapped reads might be too sub-qualified to assemble. In addition, considering the high homology showed in Didymella arachidicola isolates at the intra-specific level, we think the lower mapping rate of isolate LY268 may be related to the quality of sequencing rather than sequence differences. Our study therefore mainly focuses on the comprehensive analysis of the “called-out” variations and annotated functional genes based on the mapped reads. Nevertheless, to the point, the reviewer’s comments are very constructive. We think a comparative genomics analysis based on the complete genome assembly of several representative isolates will benefit for dealing with the reads mapping issue, and so could better investigate and illustrate the ‘gene gain’ and ‘gene loss’ in fungal pathogens in our further studies.

Point 3: A more complete picture of the structural variants present, particularly the tandem duplicated gene clusters that no doubt exist, could have been obtained by doing full PacBio assembly on several strategically-chosen isolates. Using short-read sequencing there are approaches to look at read depth variation, split-reads and the like to get an estimate of the structural variation present. But these approaches yield many false positives and are never as effective as long-read approaches.

Response 3: Thanks, this is important. We performed PacBio whole-genome assembly and Hi-C analysis to generate the complete genome for isolate YY187 as the reference genome. The clean reads of re-sequenced genomes by next-generation sequencing were all mapped to the reference with a high mapping rate around of 97%~99%, and we think the results are reliable for the present analysis in our study. Nevertheless, we do agree with the reviewer’s comment that long-read sequencing will be more effective and consider performing full PacBio assembly on several representative isolates in our future research.

Point 4: I think this paper is a nice work overlll, but I would very much like to see an analysis of the assembled unmapped reads for the isolates, and a paragraph about what this study has likely missed by not doing at least 2 full PacBio assemblies in the Discussion before it is published.

Response 4: We appreciate the reviewer’s concern. We agree that it is interesting and necessary to assemble and analyze the unmapped reads of the isolates. However, based on the phylogeny analysis, the isolate with a lower mapping rate of clean reads showed no correlations with the lesion phenotype or geographic locations, which are not the focus of this study. As the reviewer said, a comparison of full PacBio assemblies of some other strategically-chosen isolates could be more comprehensive. We have added additional description about what we will miss by not doing full PacBio assemblies for other isolates in the second paragraph of the Discussion section.

Point 5: Minor issue:
Supplemental Table S2: In the column 'Clean Reads', the commas in the numbers are in the wrong places for HuB260, LY128, NY206, SD257 making the numbers look huge at first glance.

Response 5: We appreciate the reviewer pointing this out. We have revised this to make it more accurate.

Round 2

Reviewer 2 Report

The technical quality of the manuscript has been improved compared to the previous version. The comments and questions were addressed in the revision.

Author Response

We would like to appreciate the reviewer’s two rounds of review deeply. The comments and suggestions improved our manuscript considerably.

Reviewer 3 Report

I would like to thank the reviewer's as they have gone through all the comments and suggestions carefully.

I think that the manuscript could be accepted for publication.

Author Response

We are grateful for the suggestions raised by the reviewer and checked the whole MS throughout.